# Actual, Personalized Approaches to Preserve Cognitive Functions in Brain Metastases Breast Cancer Patients

**DOI:** 10.3390/cancers14133119

**Published:** 2022-06-25

**Authors:** Monika Konopka-Filippow, Dominika Hempel, Ewa Sierko

**Affiliations:** 1Department of Oncology, Medical University of Bialystok, 15-274 Bialystok, Poland; mkonopka@onkologia.bialystok.pl (M.K.-F.); dhempel@onkologia.bialystok.pl (D.H.); 2Department of Radiotherapy I, Maria Sklodowska-Curie Bialystok Oncology Centre, 15-027 Bialystok, Poland

**Keywords:** breast cancer, brain metastases, radiotherapy, targeted therapy, cognitive function, hippocampus, quality of life

## Abstract

**Simple Summary:**

Breast cancer is the one of most common causes of brain metastases among solid malignancies, being responsible for 10–16% of all brain metastases in oncological patients. Brain metastases in the course of breast cancer significantly worsen quality of life of patients, especially in the aspect of neurocognitive domains. The review aims to summarize and integrate the current knowledge about breast cancer brain metastases, focusing on indications of certain types of treatment, and with special attention to the role of hippocampus sparing in preserving neurocognitive functions in irradiated patients.

**Abstract:**

Breast cancer (BC) is the most often diagnosed cancer among women worldwide and second most common cause of brain metastases (BMs) among solid malignancies being responsible for 10–16% of all BMs in oncological patients. Moreover, BMs are associated with worse prognosis than systemic metastases. The quality of life (QoL) among brain metastases breast cancer (BMBC) patients is significantly influenced by cognitive functions. Cancer-related cognitive deficits and the underlying neural deficits in BMBC patients can be caused via BMs per se, chemotherapy administration, brain irradiation, postmenopausal status, or comorbidities. Brain RT often leads to cognitive function impairment by damage of neural progenitor cells of the hippocampus and hence decreased QoL. Sparing the hippocampal region of the brain during RT provides protective covering of the centrally located hippocampi according to the patient’s clinical requirements. This article discusses the personalized strategies for treatment options to protect cognitive functions in BMBC patients, with special emphasis on the innovative techniques of radiation therapy.

## 1. Introduction

Breast cancer (BC) is the most often diagnosed cancer among women worldwide and accounts for over 1.7 million new cases annually. Breast cancer incidence in the European Union (EU) was more than 400,000 cases in 2018, whereas mortality of this malignancy was 138,000 and accounted for approximately 3.6% of all deaths in women and 1.8% of all deaths in the EU [1]. The mortality of BC in Europe has been declining over the last three decades, so that the 5-year survival probability due to BC in EU increased over 90% [2,3]. Unfortunately, BC is the second most common cause of brain metastases (BMs) among solid malignancies, being responsible for 10–16% of all brain metastases in oncological patients [4]. Over the last two decades, BMs have been documented in approximately one-quarter of all BC patients in the course of the disease [5]. A steady increase in newly diagnosed BMs in the course of the BC disease in patients presenting neurological symptoms results from better access to magnetic resonance imaging (MRI) as well as prolonged survival of BC patients receiving effective systemic treatment [6,7]. The results of the large multi-institutional trial, which included BC patients, revealed that the 10-year incidence of BMs was 5.2% and was associated with increased morbidity and mortality of patients. To date, standard treatment of BMs is local therapy, which includes different options, such as: surgery, whole brain radiation therapy (WBRT), and/or stereotactic radiosurgery (SRS)/stereotactic radiotherapy (SRT). Unfortunately, brain irradiation, which is one of the most frequently used modalities to treat BMs, mainly deteriorates patients’ cognitive functions by affecting the hippocampus area. The quality of life (QoL) constitutes an important issue in oncological patients. The advanced techniques and pharmacological factors directed to organ preservation should be regarded during treatment decision-making process to improve patients’ neurological and psychological condition, thus protecting the well-being of treated patients [8,9]. Additionally, advances in systemic targeted therapy decreased the urgency of local treatment or allowed for its omission in some brain metastatic breast cancer (BMBC) patients. That is why the correct treatment qualification is of paramount importance and should be based on the patient’s performance status, molecular factors, and perspectives of the anticancer systemic treatment as well as the number, size, and localization of brain lesions [10,11]. 

The review aims to summarize and integrate the current knowledge about BMBC, focusing on indications of certain types of treatment with special attention to the role of hippocampus sparing in preserving neurocognitive functions in irradiated patients. 

## 2. Brain Metastatic Breast Cancer (BMBC)—Incidence, Prognostic Factors, Qualification to Treatment

There is growing evidence that frequency of occurrence of BMs in BC patients strictly depends on molecular factors. The incidence of BMs among nonmetastatic BC patients as the site of first recurrence per year of 10 year follow-up ranged from 0.1% to 3.2%, whereas it was particularly high among metastatic HER2-overexpressing (HER2^+^) and triple negative (TN) BC patients, ranging between 22 and 36% for the former and 15 and 37% for the latter in the absence of brain screening [12]. The risk of BMs in BC patients treated with a breast-conserving approach followed by systemic treatment at 10 years was 0.7% for low or intermediate grade HR-positive HER2-negative (HR+/HER2-) subtype, 12% for high grade HR+/HER2- subtype, 8% for HR-positive HER2-positive (HR+/HER2+) subtype, 12% for HR-negative HER2-positive (HR-/HER2+) subtype, and 7% for TNBC patients [13]. Patients with TNBC have an increased risk of BMs as the first site of relapse with ranges between 25% and 45%, with a 5-year cumulative incidence of BMs of 3%, 5%, and 10% for stage I, II, and III disease, respectively [14].

Barnholtz-Sloan, Sloan et al. [15] reported that the BMs incidence rate in BC patients was 5.1% and another 14.2% of BC patients developed BMs during the treatment of disseminated-to-other-organs disease. The BMBC is recognized in 14–38% of HER2+ and TNBC patients whereas it is recognized in less than 10% of the luminal-type patients [16,17]. Similarly, the time to brain progression in radically treated patients also correlates with molecular BC status. Namely, the shortest interval from early stage BC to BMs development was documented for TNBC (22 months), and it was longer for HER2+ patients (30 months), whereas the longest time to progression was reported for HR+/HER2- BC patients (63.5 months) [18,19]. Due to the high incidence of BMs among patients with metastatic HER2+ and triple negative BC, screening for asymptomatic BMs can currently be justified [11,12].

Molecular studies have confirmed that BC subtypes with worse prognosis (higher histologic grade, unfavorable hormonal status, or HER2+ and TN BC) and thus greater risk of BMs occurrence are more often found in younger (<40 years old) rather than older women (Table 1) [20,21]. Younger BC patients (2–7% of all BC patients) are occupationally and socially active so QoL is an especially important issue for them. Importantly, younger patients with BMs have a longer life expectancy with potentially increased risk of progression of BMs [19,20].

The overall survival (OS) for BMBC patients was independently associated with subtypes and estimated above 7.1 months for HR+/HER2-, 18.9 months for HR+/HER2+, 13.1 months for HER2+, and 4.4 months for TN BC [22]. In multicenter retrospective analysis, Niikura, Hayashi, et al. [23] described OS in BMBC patients for each cancer subtype: luminal-type BMBC—9.3 months, luminal-HER2 type—16.5 months, HER2 type—11.5 months, and TNBC type—only 4.9 months (Table 2).

Helpfully, there are special scales for assessment of prognosis for BMBC patients, such as the Diagnosis-Specific Graded Prognostic Assessment (GPA) model and its modified index—breast-GPA [24]. The GPA for BC patients is based on number of BMs, age, molecular subtype, Karnofsky Performance Status (KPS), and presence of extracranial metastases (Table 1) [23,24]. Importantly, the number of BMs (>3 vs. ≤3) is the highest independent prognostic factor, besides age, tumor subtype, and KPS [25,26,27].

**Table 1 cancers-14-03119-t001:** Diagnosis-Specific Graded Prognostic Assessment (GPA) modified for breast cancer patients [23,24,26].

Prognostic Factor	GPA				
	0	0.5	1.0	1.5	2.0
**KPS**	≤60	70–80	90–100	NA	NA
**Age (years)**	≥60	<60	NA	NA	NA
**Number of BMs**	≥2	1	NA	NA	NA
**ECM**	Present	Absent	NA	NA	NA
**Subtype**	Basal	Luminal A	NA	HER2/ Luminal B	NA
**GPA Score**	0–1	1.5–2.0	2.5–3.0	3.5–4.0	
**Median Survival**	6 months	13 months	24 months	36 months	

Abbreviations: KPS—Karnofsky Performance Status; BMs—brain metastases; ECM—extracranial metastases; HER2—Human epidermal growth factor receptor 2.

**Table 2 cancers-14-03119-t002:** Median survival (MS) of brain metastatic breast cancer patients, including biological subtypes qualified to radiotherapy according to breast GPA (Graded Prognostic Assessment).

Tumor Subtype, MS (mo)	Sperduto, Mesko et al., 2020 [26]	Znidaric, Gugic et al., 2019 [27]	Darlix, Louvel et al., 2019 [22]	Niikura, Hayashi et al., 2014 [23]
Luminal A (HR-pos, HER2-neg)	14	7.1	-	9.3
Luminal B (HR/HER2-pos)	27	12.1	7.1–18.9	16.5
HER 2 (HR-neg, HER2-pos)	25	3.9	13.1–16.5	11.5
Basal like (HR/HER2-neg)	9	3.1	4.4–4.9	4.9

Abbreviation: MS—median survival, mo—months, HER2—human epidermal receptor 2, HR—hormone receptor (estrogen or progesterone receptors).

Because of the blood–brain barrier (BBB), classical systemic therapies have limited clinical benefit for BMs patients; thus, to date, local therapy has been a standard treatment. The decision of implementation surgery or certain techniques of irradiation depend on factors associated with the patient (performance status, preferences) as well as cancer-related factors (molecular subtypes, perspectives of the anticancer systemic treatment, number, size, and localization of brain lesions) [6,11]. Whole brain radiotherapy is dedicated to patients with many BMs. Stereotactic radiosurgery with or without WBRT should be considered for local disease control in patients with oligometastatic disease, defined as one to three/four BM. The treatment algorithm for BMBC patients is presented in Table 3. Median survival of BMBC patients considering various treatments is presented in Table 4.

## 3. Cognitive Functions in Brain Metastatic Breast Cancer (BMBC) Patients

### 3.1. Hippocampus

Altered cognitive functions in cancer patients can result directly from brain infiltration of the malignancy or be a distressing side effect of cancer treatment. The patients diagnosed with BC often report problems with memory, concentration, and other cognitive disabilities that can pose significant barriers to their well-being. Cancer-related cognitive deficits and the underlying neural deficits in BMBC patients can be caused via BMs, per se chemotherapy administration, brain irradiation, postmenopausal status, or comorbidities [28,29]. Brain RT often leads to cognitive function impairment. The main structure responsible for cognitive functions is the hippocampus. It is a paired brain structure of the limbic system situated in the medial temporal lobes of the telencephalon. The neural progenitor cells located in the subventricular zone as well as the hippocampus are very sensitive to radiation, so even doses of 2 Gy could be toxic. Preliminary findings suggest that irradiated neural stem cells in the hippocampus undergo apoptosis, resulting in the deterioration of cognitive functions (Figure 1) [30,31]. 

The damage of neural progenitor cells of the hippocampus after brain irradiation leads to neurocognitive deterioration and decreased QoL. Sparing the hippocampal region of the brain during RT provides protective covering of the centrally located hippocampi according to the patient’s clinical requirements. Results from the RTOG 0933 study have shown that hippocampal sparing (HS) WBRT could be manifested in a decreased rate of neurocognitive impairment after brain irradiation compared to WBRT without HS [28,29,30]. 

### 3.2. Hippocampus as ‘Organ at Risk’

During planning brain RT, many anatomical, critical structures, known as ‘Organs at Risk’ (OAR), must be taken into consideration. These include, among others, lenses, optic nerves and chiasm, cochlea, hippocampus, and brainstem. The relevant quantitative analyses of normal tissue effect in the clinic (QUANTEC) were used to compare the radiosensitivity (based α/β index) and possible side effects after ionizing radiation delivery. α/β index for the brain, the brainstem, the optic nerves, and the chiasm is 2, but this is not explicitly specified for the hippocampus [32,33]. Some investigators take assessed α/β ratio of between 2 and 3 for the hippocampus [34]. However, within the hippocampus, in the area of the dentate gyrus, there is a cluster of neural stem cells (NSCs) grouped in two niches: the subventricular zone (SVZ) and the subgranular zone (SGZ). Therefore, other authors use the α/β value of 10 for hippocampal NSCs, the same as for stem cells [35]. The preclinical experiments have demonstrated that doses of even 2 Gy cause apoptosis in NSCs [36], thereby reducing the survival of these cells by even 50%. It was shown that irradiation of the hippocampus area with doses close to 30 Gy and higher, given in conventional fractionation, leads to a decrease in NSCs proliferation rate by 93–96% after 48 h [37]. Therefore, the presence of NSCs within the hippocampus results in this structure being more sensitive to ionizing radiation than other organs in the brain.

### 3.3. The Tolerance Doses of Hippocampus during Brain RT

The dentate gyrus of the hippocampus is the most important area responsible for memory function maintaining. Gondi, Pugh et al. [30] revealed that the D _40%_ exceeding 7.3 Gy delivered to the bilateral hippocampi was associated with a decrease in delayed recall cognitive function at 18 months. Another study demonstrated that radiation doses over 40 Gy resulted in a significant atrophy of the hippocampus, which was visualized on CT [38]. Based on these data, it is proposed to spare only dentate gyrus of the hippocampus to protect memory functions after brain RT. If possible, the dose to the hippocampi (contoured as dentate gyrus) should follow the ALARA rule and preferably the *D* _40%_ of both hippocampi should be kept below 7.3 Gy and the D_max_ should not exceed 16 Gy [30,36].

### 3.4. The Side Effect of Radiation on the Hippocampus

Irradiation of the brain, particularly the area of the hippocampus, leads to cognitive deficits, which decreases patients’ quality of life. The cognitive functions relate to the thought processes used to process information coming from the outside world into the mind and contain basic aspects such as memory, attention, and association as well as complex ones, including thinking and imagination [39,40].

The most frequently described deficits of cognitive functions after brain irradiation are losses in short-term memory (less frequently—in delayed memory) and problems with information recall and learning [41]. Verbal memory disorders, necessary to understand reading text, as well as inhibition of the higher cognitive processes, necessary to behave in new and difficult situations, were also described.

The deficits in cognitive functions appear approximately two months after the brain irradiation, and the peak of their intensity falls around the fourth month [42]. Importantly, the consequences of NSC apoptosis are irreversible and usually progressive over time.

### 3.5. Hippocampal Metastases

Reliable data about frequency of metastases in the hippocampus area are indispensable to incorporate hippocampus sparing procedures into daily clinical practice in oncological patients. Fortunately, there are studies that have investigated this issue. The biological subtype of BC has previously been reported as distinct patterns associated with distant metastases. Wu, Sun et al. [43] found that the patients with TNBC were more likely to develop BMs. The proportion of patients with more than 10 BMs was higher in the group of HER2+ and TNBC patients relative to HR+/HER2− patients. In addition, patients with more than 10 BMs had a significantly increased risk of perihippocampal metastases (PHMs)—in or within 5 mm around the hippocampus (HMs), although no association was found between biological subtypes of BC and hippocampal metastases (HMs). There were 513 BMs identified in 73 HER2+ BC patients in a study by Witt, Pluard et al. [44]. The proportion of patients with HMs and PHMs was only 6.8% (5/73) and 15.1% (11/73), respectively. Another author did not found a correlation between BC biological subtype and PHMs. Therefore, even though there was a relationship between the BC biological subtype and the number of BMs [45], molecular characteristics of the primary tumor do not determine the localization of BMs [16,18,19,22]. Table 5 and Table 6 show the literature review relative to the incidence of HMs and PHMs in BC patients.

The most frequent (above 26%) anatomical locations/distribution of BMs in BC patients are the frontal lobe and cerebellum [43]. BMs occurred rarely in the hippocampus with the rate near 2–3% of all BMs. In the group of 314 BC patients with 1678 BMs, hippocampal metastases were found in 1.2% of metastases and in 4.1% of cases, but perihippocampal metastases comprised 3.5% of lesions in 11.1% of patients [46]. In another study, a potential risk of HMs and PHMs in 192 BC patients was estimated with a total of 1.356 BMs lesions relative to other malignant tumors. The frequency of HMs and PHMs was only 3.6% and 7.3%, respectively, and they occurred rarely in BC patients in contrast to, e.g., lung cancer patients [46]. Sun, Huang et al. [45] found that patients with more than 10 BM had a significantly increased risk of PHMs. Moreover, the authors reported that only the number of BMs was associated with increased risk of PHMs occurrence. The probability of PHMs significantly increased with an increase in the BM number in BC patients [45,47]. In BC patients with more than four BMs, PHMs occurred approximately four times as often relative to patients with 1–3 BMs and approximately 11 times as often in patients with more than 10 BMs vs. 1–3 BMs [46]. This was similarly reported in study by Wu, Rao et al. [46]—BC patients with a higher number of BMs (especially more than four) have a higher risk of PHMs [45]. The retrospective analysis on 565 BMs in 116 cancer patients revealed that BC patients with oligo (1–3) BMs have a lower risk of HMs and PHMs compared to patients with other cancers, e.g., lung or colorectal cancer [46]. In sum, the probability of hippocampal metastasis was low or very low in BC patients.

## 4. Treatment Options for BMBC Patients with Cognitive Function Preservation

### 4.1. Whole Brain Radiotherapy (WBRT)

The role of WBRT in BMs patients has been reconstructed during the last decade and now is dedicated mainly to the patients with multiple BMs or to patients with a small number of metastases but with unfavorable prognostic factors, as well as for patients with poor performance status (but still enough to be treated). WBRT is associated with subsequent neurotoxicity, including significant deterioration in cognitive functions and hence quality of life (QoL) due to postirradiation injury of sensitive hippocampus, especially neural stem cells in the perihippocampal region [30,31,35]. The main clinical manifestation of the hippocampal injury secondary to ionizing radiation is deterioration in memory and spatial navigation. A decrease in cognitive functions has previously been observed in BC patients receiving chemotherapy [48,49], which related to a reduction in hippocampal volume and verbal memory performance [50]. Therefore, a combination of WBRT and systemic chemotherapy may increase the risk of neurocognitive deterioration in BC patients. 

### 4.2. Hippocampal Sparing—Whole Brain Radiotherapy (HS-WBRT)

The Radiation Therapy Oncology Group (RTOG) 0933 study reported that hippocampal sparing WBRT (HS-WBRT) was protective against the adverse neurocognitive outcomes and was associated with an improved QoL of patients [30,36,51]. Moreover, the potential risk of PHMs recurrence was 4.6% for WBRT and 6.8% for sub-therapeutic irradiation in the PH region during WBRT; thus, HS-WBRT was considered safe and suitable for BC patients [30,51]. However, this study was limited by a small sample size of BC patients (No 56) [30]. Importantly, HS-WBRT in BCBM patients has a low risk of metastases and recurrence at the hippocampal avoidance region, which was reported by Sun et al. [45]. Moreover, the benefits of HS-WBRT in terms of neurocognitive domain have recently been validated [51]. The phase III randomized trial by NRG Oncology CC001 (NCT02360215) assessed patients with BMs treated with either HS-WBRT plus memantine or WBRT plus memantine. The results showed that there is lower risk of neurocognitive failure in BMs patients after HS-WBRT plus memantine compared with WBRT plus memantine (adjusted hazard ratio, 0.74; 95% CI, 0.58–0.95; *p* = 0.02). Furthermore, there were no differences in OS, intracranial progression-free survival (PFS), or neurotoxicity [51,52]. 

The low frequency of metastases within the hippocampus could potentially define this organ as a dose-limiting structure for WBRT. Dosimetric results suggest that it is now technically feasible to implement HS-WBRT [50,53]. 

### 4.3. Pharmacological Neuroprotection during WBRT

To reduce the neurocognitive decline after WBRT, different pharmacological approaches have been investigated. Memantine, an N-methyl-D-aspartate receptor antagonist, has been proved to reduce the neurocognitive dysfunctions after WBRT. Brain irradiation, especially neural stem cells of hippocampi, leads to cognitive deterioration as a result of the radiation-induced accelerated neurosclerosis, microangiopathy, and infarction [54,55]. Study of RTOG 0614 revealed that memantine as a pharmacological neuroprotector was well tolerated by patients and resulted in good neurocognitive function preservation in patients receiving WBRT with memantine compared to only WBRT at 24 weeks observation follow-up (*p* = 0.0041) [52].

The randomized phase III trial revealed that donepezil, a neurotransmitter modulator, did not significantly improve the overall composite neurocognitive domain in patients with primary brain tumors or BMs after brain RT but resulted in modest improvements in several cognitive functions, especially among patients with greater pretreatment impairments [56]. 

### 4.4. Delaying or Omitting WBRT

To reduce the neurocognitive sequelae after WBRT, there is a tendency to avoid this technique in BC patients and propose other, more organ sparing options, for instance surgical resection, SRS, SRT, or intensity modulated radiotherapy with hippocampal sparing, which is presented in Figure 2, or pharmacological approaches.

In certain groups of BMBC patients, delaying WBRT can be considered to prevent neurocognitive decline prior to the observation or another anticancer modality without differences in OS. The median OS in BMBC patients after WBRT or SRS was 4–6 months and up to 16 months if solitary BM could be removed surgically [57]. The longest OS in BMBC can be observed after SRS or SRT in HER2 positive patients in good performance status with single BM (Table 4).

Actually, WBRT alone should be considered in patients with single or multiple BMs not able to have surgery or radiosurgery with a reasonable prognosis or with an urgent need of symptom relief. For patients with a very a priori poor prognosis, best supportive care may be considered [58].

### 4.5. Surgical Resection

Surgical resection is a treatment option in large, single (diameter > 3 cm) BMs, causing raised intracranial pressure (ICP) or neurological deficits when located in eloquent brain regions or in the case of diagnostic uncertainty. Postsurgical histopathological examination allows for the confirmation of BMs with reassessment of the immunophenotype. The survival benefit of surgical resection seems to be limited to the subgroup of patients with controlled systemic disease and good performance status. After brain surgery, focal irradiation of the surgical cavity may be suggested to reduce the risk of local relapse [59,60]. Radiosurgery of the resection cavity may offer comparable survival and local control as postoperative whole-brain radiotherapy (WBRT).

### 4.6. Stereotactic Radiotherapy Vs. WBRT

WBRT plus SRS significantly decreases brain tumor recurrence rate but does not improve the survival for patients with one to four BMs [61]. The meta-analysis evaluating WBRT versus WBRT plus SRS showed that there was no difference in OS between those two groups (*p* = 0.24), but WBRT plus SRS has better local control (hazard ratio 2.88; 95% CI, 1.63–5.08; *p* = 0.0003) [62]. Moreover, in patients with one to three BMs, the OS did not differ between SRS alone and SRS plus WBRT groups (hazard ratio, 1.02; 95% CI, 0.75–1.38; *p* = 0.92) [63]. It was noted that the time to intracranial failure was significantly shorter for the SRS alone group compared with the SRS plus WBRT group (hazard ratio, 3.6; 95% CI 2.2–5.9; *p* < 0.001) (NCT00377156) [64]. Importantly, the combination of WBRT and SRS is associated with later neurocognitive dysfunction, which was proved in a randomized controlled trial. The significantly decreased learning and memory functions was observed at four months after SRS plus WBRT management compared with the SRS alone group [65]. The results of another study showed that the incidence of cognitive deterioration was lower in the SRS alone group both 3 and 12 months after treatment compared with the SRS plus WBRT group 45.5% (SRS alone) versus 94.1% (SRS plus WBRT). Authors concluded that SRS alone is a better treatment option for patients with one to three BMs than SRS plus WBRT (NCT00377156) [64], because it allows for cognitive function preservation. 

The neurocognitive function sparing during brain radiotherapy is more complicated in breast cancer patients with more than four or five BMs. Recent studies showed that omitting WBRT and application of only SRS in BC patients with 4–15 BMs can reduce the risk of neurocognitive deterioration. Yamamoto, Serizawa et al. [66] reported that there was no difference in the median OS between patients with 2–4 BMs and 5–10 BMs (hazard ratio, 0.97; 95% CI, 0.81–1.18 (less than noninferiority margin); *p* = 0.78; *p* for noninferiority < 0.0001) treated with SRS (10% of such patients had breast cancer). As before, SRS is the standard treatment in patients with limited intracranial disease or surgically inaccessible tumors (in patients with disease-controlled periphery), increasing the median survival time of BCBM patients to more than 1 year [63]. Therefore, SRS without WBRT could be an alternative treatment option for patients with five to ten BMs. Interestingly, the long-term follow-up of Yamamoto, Serizawa et al. [67]’s study revealed that the neurocognitive function did not differ between those two groups. The ongoing study, NCT03075072, investigated quality of life in patients with 5–20 BMs treated with HS-WBRT or SRS and revealed that minimizing hippocampal dose while providing tumor coverage was feasible and might translate to neurocognitive protection [68].

### 4.7. Unable to Spare the Hippocampus during Brain RT

In certain clinical situations (e.g., extensive brain tumors infiltrating the hippocampus or metastases within the hippocampus) protection of the hippocampus is impossible. The priority is to properly cover the clinical target volume with the desired radiation dose. The possibility of the avoidance of the contralateral hippocampus should be taken into account [69,70]. Furthermore, SRT or FSRT, which spare the hippocampus, instead of WBRT, should be considered. Importantly, the use of pharmacological neuroprotectors, such as memantine, is recommended as well.

### 4.8. Targeted Therapy and Delayed Brain Irradiation

The management of intracranial disease in BC patients, particularly when extracranial disease is controlled, may offer novel systemic agents with antitumor effects in the central nervous system [71]. The standard first-line treatment for metastatic HER2+ BC patients is the combination of trastuzumab, pertuzumab, and taxane based on the CLEOPATRA trial [72]. The anticancer therapy with HER2-targeted agents, such as trastuzumab, as well as chemotherapy, have been associated with significant improvements in OS following the diagnosis of BMs. The analysis of the EMILIA trial devoted to BC patients with asymptomatic BMs demonstrated that 45 BC patients treated with ado-trastuzumab emtansine (T-DM1) compared to 50 BC patients treated with combination of lapatinib and capecitabine had similar rates of cerebral progression. Moreover, the rate of cerebral progression in patients without baseline BMs was 2% versus 0.7%, respectively, and 22.2% versus 16% for those with baseline BMs [73]. The development of novel HER2-targeted therapies, such as antibody-drug conjugates (tucatinib, ado-trastuzumab emtansine, trastuzumab deruxtecan and neratinib) and small-molecule tyrosine kinase inhibitors (TKIs), have provided new therapeutic options for the management of HER2+ BMBC patients with early promising results. Patients with small, subcentimeter asymptomatic metastases may more benefit from treatment with systemic therapy and delayed radiation. In summary, in specific clinical situations, such as a patient with newly diagnosed HER2+ asymptomatic BMBC, who may be offered delayed brain irradiation (SRS or HS-WBRT), this is an exception to standards of care [74]. The paradigm for treatment patients with HER2+ BMBC is evolving and will continue to do so as novel agents move into earlier lines of the treatment algorithm [75]. It is obvious that avoidance or delaying of brain irradiation in BC patients should positively influence their neurological and psychological condition.

## 5. Conclusions

Brain metastases in the course of breast cancer significantly worsen quality of life of patients, especially in the aspect of neurocognitive domains. Moreover, BMs are associated with a worse prognosis than systemic metastases. A neurological condition and satisfactory QoL is extremely important for BC patients. This article draws the following conclusions: firstly, the possibility of BMs in the region of the hippocampus and in the perihippocampal area in BC patients might be mainly related to the biological subtype of BC and the number of BMs. Secondly, the probability of hippocampal metastasis in BMBC patients was low. Thirdly, there are many treatment strategies for BMs to consider, including surgery and RT with modalities, which enable the protection of cognitive functions and in consequence QoL.

## Figures and Tables

**Figure 1 cancers-14-03119-f001:**
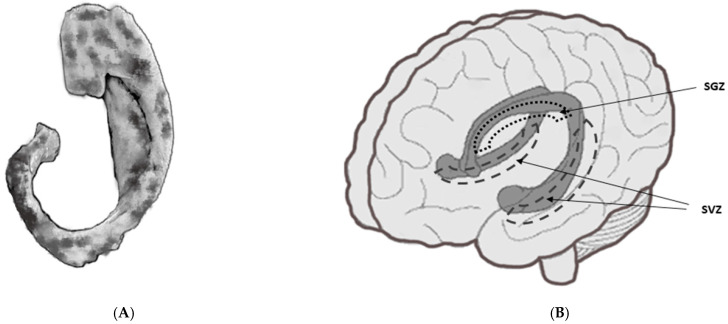
The sketch of hippocampus (**A**) and localization of neural progenitor cells of hippocampus (**B**). SGZ—subgranular zone, SVZ—subventricular zone.

**Figure 2 cancers-14-03119-f002:**
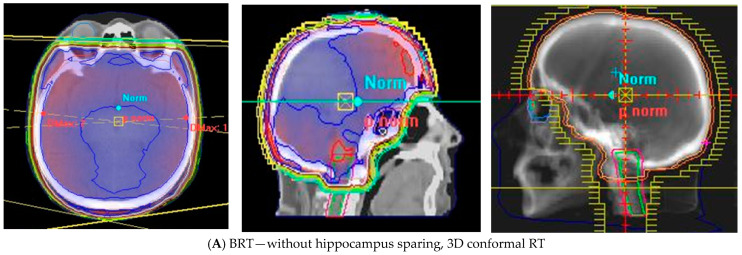
Brain radiotherapy in patient with brain metastasis without hippocampus sparing (**A**), with sparing hippocampus by HS-WBRT (**B**) or FSRT (**C**); Abbreviations: RT—radiotherapy, WBRT —whole brain radiotherapy, VMAT—volumetric arc therapy, HS-WBRT—hippocampal sparing whole brain radiotherapy, FSRT—fractionated stereotactic radiotherapy.

**Table 3 cancers-14-03119-t003:** Simple treatment algorithm for the approach to a BC patient affected by BM.

**Single Brain Metastases (1–4)**		
**Surgical Management**	**SRS/SRT**	**WBRT/BSC**
- KPS > 70 - Solitary lesion - Large diameter > 3 cm - Mass effect - Cerebellum localization - Need for histology verification	- KPS > 70 - Stable disease - Localization in deep brain area/eloquent area	- KPS < 70 - Preference of patient
**Multiple brain metastases (more than 4)**		
**Surgical management**	**SRS/SRT**	**WBRT/ BSC**
- Large diameter of dominant lesion - KPS > 70 - Need for histology verification	- KPS >70	- KPS < 70 - SRS/SRT cannot be safely used - Preference of patient - Instable disease, rapid progression

Abbreviation: KPS—Karnofsky Performance Status, WBRT—whole brain radiation therapy, SRS—Stereotactic radiosurgery, SRT—Stereotactic radiotherapy, BSC—best supportive care.

**Table 4 cancers-14-03119-t004:** Median survival (MS) of brain metastatic breast cancer patients, including various modalities.

Treatment Options in BM BC Patients	WBRT	SRS	WBRT + SRS	S + SRS	S + WBRT	S + WBRT + SRS	WBRT + SRS
MS (mo)	13	16	15	19	25	24	16

Abbreviation: MS—median survival, mo—months, WBRT—Whole Brain Radiation Therapy, SRS—stereotactic radiosurgery, S—surgery, BC- breast cancer, BM—brain metastases.

**Table 5 cancers-14-03119-t005:** The incidence of hippocampal and perihippocampal metastases in breast cancer patients.

Study, Year	No	Incidence of Brain Metastases	
		HMs No (%)	PHMs No (%)
**Wu, Sun et al., 2016** [43]	192	7 (3.6%)	14 (7.3%)
**Witt, Pluard et al., 2014** [44]	73	6.8%	15%
**Sun, Huang et al., 2016** [45]	314	2 (4.1%)	5 (11.1%)
**Wu, Rao et al., 2015** [46]	56	4.1%	5.5%

Abbreviations: BC—breast cancer, HM—hippocampal metastases, PHM—perihippocampal metastases, No—Number of patients.

**Table 6 cancers-14-03119-t006:** The incidence of hippocampal and perihippocampal metastases in breast cancer patients estimated by Han, Cai et al. [47].

Han, Cai et al., 2017 [47]	No	HM < 5 mm		PHM < 10 mm		PHM < 20 mm	
	45	No	Yes	No	Yes	No	Yes
Luminal	17	16/94.1	1/5.9	15/88.2	2/11.8	15/88.2	2/11.8
HER 2 over-expressed	13	12/92.3	1/7.7	11/84.6	2/14.5	11/84.6	2/15.4
Triple-negative	13	13/ 100	0/0	12/92.3	1/7.7	11//84.6	2/15.4

Abbreviations: BC—breast cancer, HER—human epidermal receptor, HM—hippocampal metastases, PHM—perihippocampal metastases, No—Number of patients.

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
