# Peer review of "Actual, Personalized Approaches to Preserve Cognitive Functions in Brain Metastases Breast Cancer Patients"

_cancers, 2022, doi:10.3390/cancers14133119_

Round 1

Reviewer 1 Report

Thank you for the opportunity to review this manuscript.

Here, the authors provide a review on the treatment of breat cancer metastasis, emphasizing the role of hippocampal sparing during radiotherapy.

The study is original, well written, brief, and concise. It offers updated information for oncologists, neurosurgeons, and radiotherapists alike.

Layout and format: The manuscript is structured and meets the expected format of the targeted journal. 

Title: The title of the manuscript is reflecting the content of the article.

Abstract: The abstract is well-structured and reflects the content of the article. 

Introduction: The introduction describes the aim of the study accurately. 

Methods and statistics: The authors describe data acquisition and the experimental design. The used statistical tests are appropriate and sufficient.

Results: The presentation of the results is clear and stringent. The significant limitations are listed.

My recommendation: Accept

Author Response

The authors would like to thank Reviewer nr 1 for the full acceptance of the text of the  manuscript.

Reviewer 2 Report

The authors, personalized approaches to preserve cognitive functions in brain metastases breast cancer patients.

1. What is the difference between the hippocampus and another brain site in terms of radiation sensitivity.

2. How should we try if we cannot conserve the hippocampus?

3. How many percentages of the hippocampus should be protected if we want to maintain memory function?

4. Please describe the side effect of the radiation on the hippocampus.

Author Response

The authors would like the thank the Reviewer nr 2 for carefull reviewing the text of the manuscript and usefull points raised. Below, please find the answers. All the information was added to the text of the manuscript.

  1. What is the difference between the hippocampus and another brain site in terms of radiation sensitivity.

During planning brain RT, many anatomical, critical structures, known as ‘Organs at Risk’ (OAR), must be taken into consideration. These include, among others, lenses, optic nerves and chiasm, cochlea, hippocampus and brainstem. The relevant quantitative analyses of normal tissue effect in the clinic (QUANTEC) were used for comparing the radiosensitivity (based α/β index) and possible side effects after ionizing radiation delivery. α/β  index for the brain, the brainstem, the optic nerves and the chiasm is 2, but this is not explicitly specified for the hippocampus [[1], [2]]. Some investigators take assessed α/β ratio of between 2 and 3 for the hippocampus [[3]]. However, within the hippocampus, in the area of the dentate gyrus, there is a cluster of neural stem cells (NSCs) grouped in two niches: the subventricular zone (SVZ) and the subgranular zone (SGZ).  Therefore other authors use α/β value of 10 for hippocampal NSCs, the same as for stem cells [[4]]. The preclinical experiments have
demonstrated that doses of even 2 Gy cause apoptosis in NSCs [[5]], thereby reduce the survival of these cells by even 50%. It was shown that irradiation of the hippocampus area with doses close to 30 Gy and higher, given in conventional fractionation, leads to a decrease in NSCs proliferation rate by 93–96% after 48 hours [[6]]. Therefore  NSCs presence within the hippocampus results that this structure is more sensitive to ionizing radiation than other organs in the brain.

  1. How should we try if we cannot conserve the hippocampus?

In certain clinical situations (e.g. extensive brain tumors infiltrating the hippocampus or metastases within the hippocampus) protection of the hippocampus is impossible. The priority is to properly cover clinical target volume with desire radiation dose. The possibility of the avoidance of the contralateral hippocampus should be taken into account  [[7], [8]]. Furthermore, SRT or FSRT witch sparing hippocampus instead WBRT should be considered. Importantly, the use of pharmacological neuroprotectors as memantine is recommended, as well.

  1. How many percentages of the hippocampus should be protected if we want to maintain memory function?

Dentate gyrus of the hippocampus is the most important area responsible for memory function maintaining. Gondi et al. [[9]] revealed that the D 40% exceeding 7.3 Gy delivered to the bilateral hippocampi was associated with a decrease in delayed recall cognitive function at 18 months. Other study demonstrated that radiation doses over 40 Gy resulted in a significant atrophy of the hippocampus, which was visualized on CT [[10]]. Based on these data it is proposed to spare only dentate gyrus of the hippocampus to protect memory functions after brain RT. If possible, the dose to the hippocampi (contoured as dentate gyrus) should follow ALARA rule and preferably the D 40% of both hippocampi should be kept below 7.3 Gy and the Dmax should not exceed 16 Gy [3].

  1. Please describe the side effect of the radiation on the hippocampus.

Irradiation of the brain, particularly the area of the hippocampus, leads to cognitive deficits, which decreases  patients’ quality of life. The cognitive functions relate to the thought processes used to process information coming from the outside world into the mind and contain basic aspects such as memory, attention, and association as well as complex ones, including thinking and imagination [[11], [12]].

The most frequently described deficits of cognitive functions after brain irradiation are losses in short-term memory (less frequently - in delayed memory), and problems with information recall and learning [[13]]. Verbal memory disorders, necessary to understand reading text, as well as inhibition of the higher cognitive processes necessary to behave in new and difficult situations were also decribed.

The deficits in cognitive functions appear approximately two months after the brain irradiation, and the peak of their intensity falls around the fourth month [[14]]. Importantly, the consequences of NSC apoptosis are irreversible and usually progressive over time.

[1] Marks, L.B.; Yorke, E. D.; Jackson, A.; Ten Haken, R. K.; Constine, L. S.; Eisbruch, A.; Bentzen, S. M.; Nam, J., & Deasy, J. O.Use of normal tissue complication probability models in the clinic. Int. J. Radiat. Oncol. Biol. Phys. 2010, 76: pp. S10-S19.

[2] Lawrence, Y. R., Li, X. A., el Naqa, I., Hahn, C. A., Marks, L. B., Merchant, T. E., & Dicker, A. P. Dose-volume effects in the brain. Int. J. Radiat. Oncol. Biol. Phys. 2010, 76: pp. S20-S27.

[3] Lambrecht, M., Eekers, D., Alapetite, C., Burnet, N. G., Calugaru, V., Coremans, I., Fossati, P., Høyer, M., Langendijk, J. A., Méndez Romero, A., Paulsen, F., Perpar, A., Renard, L., de Ruysscher, D., Timmermann, B., Vitek, P., Weber, D. C., van der Weide, H. L., Whitfield, G. A., Wiggenraad, R., et al. Work package 1 of the taskforce “European Particle Therapy Network” of ESTRO. Radiation dose constraints for organs at risk in neuro-oncology; the European Particle Therapy Network consensus. Radiother. Oncol. 2018, 128, 1:26-36. doi: 10.1016/j.radonc.2018.05.001.

[4] Marsh, J. C., Godbole, R., Diaz, A. Z., Gielda, B. T., & Turian, J. V. Sparing of the hippocampus, limbic circuit and neural stem cell compartment during partial brain radiotherapy for glioma: a dosimetric feasibility study. J Med. Imaging Radiat. Oncol. 2011; 55: 442–449.

[5] Gondi, V., Hermann, B. P., Mehta, M. P., & Tomé, W. A. Hippocampal dosimetry predicts neurocognitive function impairment after fractionated stereotactic radiotherapy for benign or low-grade adult brain tumors. Int. J. Radiat. Oncol. Biol. Phys. 2012, 83: e487–493.

[6] Mizumatsu, S., Monje, M. L., Morhardt, D. R., Rola, R., Palmer, T. D., & Fike, J. R. Extreme sensitivity of
adult neurogenesis to low doses of X-irradiation. Cancer Res. 2003, 63, 14: 4021–4027,

[7] Konopka-Filippow, M., Sierko, E., Hempel, D., Maksim, R., SamoÅ‚yk-Kogaczewska, N., Filipowski, T., Rożkowska, E., Jelski, S., Kasprowicz, B., Karbowska, E., SzymaÅ„ski, K., & Szczecina, K.The Learning Curve and Inter-Observer Variability in Contouring the Hippocampus under the Hippocampal Sparing Guidelines of Radiation Therapy Oncology Group 0933. Current oncology. 2022, 29, 4: 2564–2574. https://doi.org/10.3390/curroncol29040210

[8] Sapienza, L. G., Ludwig, M. S., Mandel, J. J., Nguyen, D. H., & Echeverria, A. E. Could patients benefit from whole-brain radiotherapy with unilateral hippocampus sparing?. Rep. Pract. Oncol. Radiother. 2021, 26, 3: 454–456. https://doi.org/10.5603/RPOR.a2021.0059

[9] Gondi V., Hermann B.P., Mehta M.P., Tomé W.A.: Hippocampal dosimetry predicts neurocognitive function impairment after fractionated stereotactic radiotherapy for benign or low-grade adult brain tumors. Int. J. Radiat. Oncol. 2013, 85: pp. 348-354.

[10] Seibert, T. M., Karunamuni, R., Bartsch, H., Kaifi, S., Krishnan, A. P., Dalia, Y., Burkeen, J., Murzin, V., Moiseenko, V., Kuperman, J., White, N. S., Brewer, J. B., Farid, N., McDonald, C. R., & Hattangadi-Gluth, J. A. Radiation dose-dependent hippocampal atrophy detected with longitudinal volumetric magnetic resonance imaging. Int. J. Radiat. Oncol. 2017, 97: pp. 263-269.

[11] Shang, W., Yao, H., Sun, Y., Mu, A., Zhu, L., & Li, X. Preventive Effect of Hippocampal Sparing on Cognitive Dysfunction of Patients Undergoing Whole-Brain Radiotherapy and Imaging Assessment of Hippocampal Volume Changes. BioMed Res. Inter. 2022, 4267673. https://doi.org/10.1155/2022/4267673

[12] Opitz, B. Memory function and the hippocampus. Front. Neurol. Neurosc. 2014, 34: 51–59. https://doi.org/10.1159/000356422

[13] Giuseppe, Z. R., Silvia, C., Eleonora, F., Gabriella, M., Marica, F., Silvia, C., Mario, B., Francesco, D., Savino, C., Milly, B., Frezza, G. P., Maurizio, Z., & Morganti, A. G. Hippocampal-sparing radiotherapy and neurocognitive impairment: A systematic literature review. J Cancer Res. Ther. 2020, 16, 6: 1215–1222. https://doi.org/10.4103/jcrt.JCRT_573_17

[14] Welzel, G., Fleckenstein, K., Schaefer, J., Hermann, B., Kraus-Tiefenbacher, U., Mai, S. K., & Wenz, F. Memory function before and after whole brain radiotherapy in patients with and without brain metastases. Int. J Radiat. Oncol. Biol. Phys. 2008, 72, 5: 1311–1318,
doi: 10.1016/j.ijrobp.2008.03.009,

Round 2

Reviewer 2 Report

The authors replied well, so the manuscript is suitable for publication.